

# Hyper-accumulation of legacy fallout radionuclides in cryoconite on Isfallsglaciären (Arctic Sweden) and their downstream distribution

Caroline C. Clason[1], Will H. Blake[1], Nick Selmes[2], Alex Taylor[1], Pascal Boeckx[3], Jessica Kitch[1], Stephanie C. Mills[4], Giovanni Baccolo[5] & Geoffrey E. Millward[1]

[1]School of Geography, Earth and Environmental Sciences, University of Plymouth, Plymouth, PL4 8AA, UK
[2]Plymouth Marine Laboratory, Plymouth, PL1 3DH, UK
[3]Isotope Bioscience Laboratory – ISOFYS, Ghent University, Ghent, Belgium
[4]School of Earth, Atmospheric and Life Sciences, University of Wollongong, Wollongong, NSW 2522, Australia
[5]Department of Environmental and Earth Sciences, University Milano-Bicocca, Milano, Italy
*Correspondence to*: Caroline C. Clason (caroline.clason@plymouth.ac.uk)

**Abstract.** The release of legacy fallout radionuclides (FRNs) in response to glacier retreat is a process that has received relatively little attention to date, yet may have important consequences as a source of secondary contamination as glaciers melt and down-waste in response to a warming climate. The prevalence of FRNs in glacier-fed catchments is poorly understood in comparison to other contaminants, yet there is now emerging evidence from multiple regions of the global cryosphere for substantially augmented FRN activities in cryoconite. Here we report concentrated FRNs in both cryoconite and proglacial sediments from the Isfallsglaciären catchment in Arctic Sweden. Activities of some FRNs in cryoconite are two orders of magnitude above those found elsewhere in the catchment, and above the activities found in other environmental matrices outside of nuclear exclusion zones. We also describe the presence of the short-lived cosmogenic radionuclide $^7$Be in cryoconite samples, highlighting the importance of meltwater-sediment interactions in radionuclide accumulation in the ice surface environment. The presence of fallout radionuclides in glaciers may have the potential to impact local environmental quality through both isolated hotspots of radioactivity caused by glacier down-wasting, and downstream transport of contaminants to the proglacial environment through interaction with sediments and meltwater. We thus recommend that future research in this field focusses on processes of accumulation of FRNs and other environmental contaminants in cryoconite, and whether these contaminants are present in quantities harmful for both local and downstream ecosystems.





## 1 Introduction

The Arctic has received considerable attention with respect to environmental change in recent decades as it faces pressures
from a changing climate and anthropogenic activities. Long-range atmospheric transport of contaminants from distal sources
is a contributor to changing environmental quality in the Arctic, particularly during positive phases of the North Atlantic
Oscillation (NAO) (Duncan and Bey, 2004; Macdonald et al., 2005; Stohl 2006), compounded by the influence of the global
distillation process which redistributes some contaminants, whose mobility is influenced by temperature, from warmer to
cooler regions (Wania and Mackay, 1995). In addition, the deposition of airborne materials onto glacier surfaces has been
shown to impact bare ice albedo through a darkening of the surface (Keegan et al., 2014; Tedstone et al., 2017), acting as a
catalyst for increased ice surface melt (Box et al., 2012), while the cryosphere has also been recognized to be an active
component within the biogeochemical cycle of some contaminants (e.g. Vorkamp & Rigét, 2014). Contaminants are deposited
onto glacier surfaces following efficient scavenging from the atmosphere by snow (Franz & Eisenreich, 1998; Herbert et al,
2006), and it has been observed that specific anthropogenic substances, such as persistent organic pollutants (POPs), are
preferentially accumulated in cold environments and glaciers (Grannas et al., 2013). As snow compacts to form firn and ice,
these legacy contaminants are accumulated within the ice column, with glaciers acting as "reservoirs" for contaminants
(Steinlin et al., 2015; Miner et al., 2018). But due to retreat and increased melt rates, glaciers are now releasing these legacy
contaminants and behaving as a secondary source (Bizzotto et al., 2009; Bogdal et al., 2009).

Within this context, cryoconite plays a peculiar and unique role. Cryoconite is a particulate matter found on the ice surface
which consists of a mixture of organic and inorganic materials, including mineral matter, black carbon, and microbial life
(Cook et al., 2016). It often accumulates within holes formed via preferential melting due to the low albedo of cryoconite in
comparison to the surrounding ice. Due to the local concentration of nutrients and the availability of seasonal liquid water and
solar radiation, cryoconite is a hotspot for microbial life on glaciers (Takeuchi et al., 2001; Zawierucha et al., 2019). Recent
studies have highlighted that cryoconite acts as an absorbent that accumulates certain materials, including potential
contaminants such as heavy metals and anthropogenic organics (e.g. Łokas et al., 2016; Baccolo et al., 2017; Li et al., 2017;
Weiland-Bräuer et al., 2017). Aided by its interaction with meltwater during the melt season, cryoconite accumulates several
atmosphere-derived materials, acting as a temporary sink before the release of these substances into the downstream proglacial
environment (Baccolo et al., 2020a). The rich microbial life which flourishes in cryoconite holes also plays an important
biogeochemical role, as it has been demonstrated that the bioavailability of carbon, nitrogen, and phosphorus in Antarctica is
increased in cryoconite due to its microbial activity (Bagshaw et al., 2013), while augmented levels of heavy metals, including
Pb, Cd, Cu and Zn, have also been found in Arctic cryoconite (Łokas et al., 2016).

Recently it has been established that cryoconite accumulates fallout radionuclides (FRNs), including products of nuclear
weapons testing and nuclear accidents, and natural radionuclides such as $^{210}$Pb and cosmogenic $^{7}$Be. (Appleby, 2008; Taylor
et al., 2019). The prevalence of FRNs in glacier-fed catchments remains poorly understood in comparison to other atmospheric



contaminants, however a small number of studies to date have reported high activity levels of FRNs in cryoconite in the
European Alps (Tieber et al., 2009; Baccolo et al., 2017; 2020b; Wilflinger et al., 2018), the Caucasus (Łokas et al., 2018),
Svalbard (Łokas et al., 2016), and Canada (Owens et al., 2019). The environmental fate of the radioactivity accumulated in
cryoconite remains uncertain, in addition to any potential socio-economic impacts linked to the release of FRNs into glaciated
catchments. Furthermore, the processes governing downstream accumulation in proglacial areas, and subsequent dilution in

the hydrological system, have not yet been explored.

To contribute to critical knowledge on the downstream transport and accumulation of FRNs in glacial catchments, we present
a comparison of radionuclide concentrations and trace elements from cryoconite and proglacial sediments in the
Isfallsglaciären catchment of Arctic Sweden, sampled during August 2017. A combination of gamma spectrometry,
wavelength-dispersive X-Ray fluorescence spectrometry, and elemental analysis of bulk stable carbon and nitrogen isotopes

was conducted to establish the activities of radionuclides and stable elements in each sample. Combining the results we report
here from Isfallsglaciären with previous observations of FRNs in cryoconite (e.g. Tieber et al., 2009; Baccolo et al., 2017;
Łokas et al., 2018; Owens et al., 2019), we argue that the accumulation of FRNs on glaciers is not limited to localised "hot
spots" near sites of accidents, but is widespread across the global cryosphere. Furthermore we demonstrate that the FRN
activities detected in cryoconite on Isfallsglaciären are considerably higher than those found in a range of proglacial sediment

settings within the catchment, highlighting both cryoconite's unique ability to accumulate FRNs efficiently, while also
demonstrating that these activity levels are some of the highest ever recorded outside of nuclear exclusion zones.

**2 Study site**

Isfallsglaciären is a small, ~1km$^2$ polythermal valley glacier in the Tarfala Valley of Arctic Sweden. It sits on the eastern flanks

of Sweden's highest mountain, Kebnekaise (2096 m a.s.l.), at 67.9°N (Fig. 1), and while thinning substantially, has roughly
maintained its terminus position since 1990, prior to which it retreated at an average rate of ~4 m/a between 1916 and 1990
(Ely et al., 2017). The glacier terminus is split into two lobes by a bedrock outcrop. Isfallsglaciären was chosen for this study
due to the closed nature of the proglacial catchment, which is constrained by the presence of large latero-frontal Holocene
moraines (Fig. 1), the innermost of which were overridden by an advance in 1916 (Karlén, 1973). The ice surface of the north

lobe is very steep and heavily crevassed, and restricted the collection of cryoconite samples in this study to the south lobe.
Glacial meltwater emerges within two braided proglacial outlets from the north and south lobes, which feed into two proglacial
lakes, Frontsjön and Isfallssjön, situated within 700m of the present-day terminus. We targeted Isfallssjön when extracting a
lake sediment core as it significantly predates Frontsjön, which formed following glacial retreat past an overdeepening in the
forefield after 1959 (Karlén, 1973), and is fed by both of the proglacial outlet streams.





**Figure 1: Overview map of the Isfallsglaciären catchment (A), sediment sampling within the proglacial zone (B), and cryoconite sampling on the surface of the southern glacier lobe (C). Imagery source: © Google Earth, 67°54'55.25"N; 18°35'32.13"E (image from 8/10/2013).**

## 3 Methods

### 3.1 Sampling strategy and sample preparation

Our sampling strategy was designed to characterize the range of sources contributing to sediment accumulation in the most distal lake, Isfallssjön, and encompasses cryoconite from the ice surface, sediments within the two proglacial outlets, surface

sediments from the overridden inner slopes of the north and south moraine, and sediments from the central foreland (Fig. 1).





We conducted cryoconite sampling in two transverse profiles to investigate the effects of aspect and distance from the valley side on the accumulation of FRNs and other materials, and sampled in the proglacial outlets at intervals along the reaches of the two streams to investigate any possible downstream changes in FRN activity concentrations. Each sample was collected in a spatially-integrated manner by sampling from five sites within a metre of a central point. We also retrieved a 38 cm lake sediment core from Isfallssjön using a HTH 90 mm diameter gravity corer, and extruded the core on-site at 1 cm intervals. Samples were subsequently oven-dried at 100°C until a constant weight, and the <75 µm component retrieved for subsequent geochemical analyses. Due to the limited amount of cryoconite available for sampling at each supraglacial site, we reserved the entire bulk sample to ensure we had sufficient material for gamma spectrometry. Lake core sections were also preserved in bulk for particle size analysis. Particle size analysis was performed in triplicate on all samples taken from the catchment, using laser diffraction. We use the surface area-weighted mean particle size, or D[3,2], for subsequent data analysis presented here as this is the most sensitive measure where fine particulates are common within the size distribution (Malvern, 2015), and relevant where reactivity and bioavailability are of potential importance.

## 3.2 Gamma spectrometry

Radioactive analyses were carried out in the ISO9001 accredited Consolidated Radio-isotope Facility (CoRIF) at the University of Plymouth, applying an established methodology (e.g. Wynants et al., 2020). Particulate samples for the well detector were packed and sealed into 4 mL plastic vials and samples for analysis on the planar detector were packed and sealed into 90 mm plastic petri dishes. Sample weighing was conducted on a calibrated balance. Samples were packed within 14 days of return to the laboratory and were incubated for 22 days prior to analysis to allow the development of secular equilibrium along the $^{238}$U decay chain. Gamma counting was conducted using well (GWL-170-15-S; N-type) and planar (GEM-FX8530-S; N-type) spectrometers, both consisting of liquid nitrogen cooled, high purity germanium semiconductor detectors (EG&G ORTEC, Wokingham, UK). The well detector had a full width-half maximum (FWHM) for the 1330 keV line of $^{60}$Co of 2.17 keV, and the planar detector a FWHM of 1.76 keV and a relative efficiency of 50.9%. The energies, peak widths, and efficiencies of the gamma spectrometers were calibrated using a natural, homogenised soil, with low background activity, which had been spiked with a certified, traceable mixed radioactive solution (80717-669 supplied by Eckert & Ziegler Analytics, Georgia, USA). Calibration relationships were derived using ORTEC GammaVision© software. After incubation, the spiked soils and the samples from Isfallsglaciären, and empty sample containers as blanks, were counted for at least 24 h, and all activities were decay-corrected to the sample collection date. The uncertainties were estimated from the counting statistics and are quoted with a 2-sigma counting error. Unsupported $^{210}$Pb ($^{210}$Pb$_{un}$) activities were obtained by the subtraction of $^{226}$Ra activity, deduced from the gamma emissions of $^{214}$Pb, from the measured total activity of $^{210}$Pb ($^{210}$Pb$_T$). Quality control analyses were carried out regularly using soils from the IAEA world-wide proficiency tests, including a moss soil (IAEA-CU-2009-03) and a soil (IAEA-TEL-2012-03) (Table A1).



### 3.3 Wavelength-dispersive X-Ray fluorescence spectrometry

We analysed all samples for a full suite of major and minor elements using wavelength-dispersive X-Ray fluorescence (WD
XRF) spectrometry. For the proglacial area and lake core, each sample was milled using a Fritsch pulverisette, mixed with a
Ceridust 6050M S1000 polypropylene wax binder (Clariant, Switzerland), and pressed into a pellet. The dried cryoconite
samples were powdered by hand using a pestle and mortar prior to being packed into 40mm diameter cups fitted with 6 μm
polypropylene spectromembrane (Chemplex, USA). All samples were packed to the same volume and left to settle for 24
hours prior to analysis. Analyses were undertaken in the CoRIF lab by WD XRF spectrometry (Axios Max, PANalytical,
Netherlands). The instrument was operated at 4 kW using a Rh target X-ray tube. During sequential analysis of elements tube
settings ranged from 25 kV, 160 mA for low atomic weight elements up to 60 kV, 66 mA for higher atomic weight elements.
All analyses were undertaken using the Omnian analysis application (PANalytical, Netherlands) under a medium of He. This
approach offers a rapid and non-destructive means of determining a wide range of elemental concentrations in cryoconite.
Repeatability of the approach was assessed by repacking and analysing cryoconite samples in triplicate with relative standard
deviation found to be <10% across triplicates. Cross comparison to results obtained from a validated inductively coupled
plasma optical emission spectrometry (ICP-OES) procedure showed XRF-derived concentrations were in close agreement
(within 15 % relative to ICP-OES) for the elements of interest.

### 3.4 Stable isotope analysis

The dried cryoconite samples were ground by hand using a pestle and mortar. Particulate N and C were determined via
elemental analysis (Carlo-Erba, EA1110, Italy). The instrument was calibrated using acetanilide, empty pre-combusted
capsules were analysed as blanks, and the accuracy of the analyses were checked using the certified reference material PACS-
2 (National Research Council of Canada). The results showed that the analyses were accurate to within 10% of certified values.
The ground samples for $^{13}C/^{12}C$ and $^{15}N/^{14}N$ analysis were packed into tin capsules and weighed using a calibrated balance.
The $^{13}C/^{12}C$ and $^{15}N/^{14}N$ ratios were determined at the Isotope Bioscience Laboratory at Ghent University using an elemental
analyser (ANCA-SL, SerCon, UK) coupled to an isotope ratio mass spectrometer (20-22, Sercon, UK). The measured $\delta^{13}C$
and $\delta^{15}N$ values were given relative to the international standards, Vienna PeeDee Belemnite (V-PDB) and Air, respectively.
This calibration was done using the IA-R001 $^{15}N/^{13}C$ wheat flour laboratory standard ($\delta^{13}C$ V-PDB = -26.43 ± 0.08 ‰ and
$\delta^{15}N$ AIR = +2.55 ± 0.22 ‰) and an in house quality assurance organic reference. The average standard deviation on the δ
value was determined by measuring five randomly selected samples in triplicate, giving a standard deviation of 0.32 ‰ for
$^{13}C$ and 0.14 ‰ for $^{15}N$.

### 3.5 Constant rate of supply modelling

The sedimentary archive from the proglacial lake core was used to construct a sedimentation chronology using fallout $^{210}Pb$
supported by known $^{137}Cs$ date horizons. Reference dates from Chernobyl (1986) and weapons testing (1963 peak and 1952
onset) were used to constrain the chronology following principles outlined in Appleby (2002) in two phases. Due to known



disruption of sediment flux from the glacier to Isfallssjön in 1959 with formation of Frontsjön, the constant rate of supply (CRS) model was applied to the core in two sections. This was done to account for any potential change in secondary $^{210}Pb_{un}$ supply. The CRS model was initially run fitting the $^{210}Pb$ profile to the lowermost measurement of $^{241}Am$, wherein $^{241}Am$ is known to have been predominantly supplied by global weapons testing fallout (Olszewski et al., 2018). The core was then split at dated horizon 1959 and the lower section analysed with a separate CRS model benchmarked to the 1952 onset of $^{137}Cs$ fallout and the 1959 minor peak linked to global fallout records. Due to low activity concentrations and detection challenges in the lowermost section, the profile tail was modelled using an exponential function fitted ($r^2 = 0.99$) to three high precision measurements derived from extended count times, noting the tail represents a small overall proportion of total inventory. Horizon dates and sediment accumulation rates were derived as outlined by Appleby (2002) for each separate model application.

## 4 Results and Discussion

### 4.1 Geochemical composition of cryoconite

Fourteen samples of cryoconite were retrieved from the surface of Isfallsglaciären, which are characterized by the range of radionuclide concentrations described in Table A2. The mean activity concentrations of $^{137}Cs$, $^{210}Pb_{un}$ and $^{241}Am$ in cryoconite are 3069 ± 941, 9777 ± 780, and 25.8 ± 16.7 Bq kg$^{-1}$ respectively, reaching a maximum of 4533 ± 350, 14663 ± 1167, and 74.0 ± 10.2 Bq kg$^{-1}$. While $^{210}Pb$ is a natural radioisotope derived from the decay of $^{222}Rn$ in the atmosphere, $^{137}Cs$ and $^{241}Am$ are anthropogenic FRNs, distributed via atmospheric transport, and are common fission by-products from nuclear reactors and weapons testing (Lindblom, 1969). The anthropogenic radionuclide $^{137}Cs$ is partially soluble in water, and is potentially hazardous to both animal and human health. With a half-life of 30.17 years it is relatively short-lived in the environment, and the $^{137}Cs$ deposited globally through long-range atmospheric deposition following the Chernobyl accident has decayed by 50% since 1986 (Olszewski et al., 2018). The half-life of $^{241}Am$ is considerably longer at 432.2 years, and is increasing in the environment due to the short half-life (14 years) of its parent radionuclide $^{241}Pu$. $^{241}Am$ is an alpha emitter and less exchangeable (acid/water soluble) than $^{137}Cs$ (Kovacheva et al., 2014), however is potentially harmful if ingested (e.g. Harrison et al., 1994). The primordial radionuclide $^{40}K$ is also detected in our cryoconite samples at relatively high activities (an average of 1839 ± 168 and a maximum of 2054 ± 207 Bq kg$^{-1}$). The $^{40}K$ mean activities found in cryoconite on Isfallsglaciären exceed the maximum activities found on both the Forni and Morteratsch glaciers in the Italian and Swiss Alps (770 ± 200 and 810 ± 55 Bq kg$^{-1}$ respectively) as reported by Baccolo et al. (2020a), and the considerably higher maximum of 1440 ± 40 Bq kg$^{-1}$ recorded on the Stubacher Sonnblickkees glacier of the Austrian Alps by Wilfinger et al. (2018). Since $^{40}K$ is a natural component of rock, the high activities found at Isfallsglaciären are likely related to the geochemical signature of the surrounding catchment geology, and in particular to the abundance of potassium in the rock and sediment.

By considering both the activities of radionuclides and the content of C and N, it is possible to explore the relationship between the organic content of cryoconite and the distribution of radioactivity. The spatial variability of activities for selected natural





and anthropogenic radionuclides is illustrated in Figure 2. The mass fraction of C and N (%C and %N) measured in the cryoconite samples through bulk stable isotope analysis is also shown in Figure 2C, illustrating the influence of organic content on accumulation of radionuclides. The relationship between organic content and accumulation of some radionuclides is illustrated by the relative low values of both %N and %C, and natural radionuclides $^{210}$Pb and $^{7}$Be, in the southernmost samples from the upper glacier transect, with higher relative values in samples collected to the north. Values of %C and %N cover a wide range of 6.39-24.51% and 0.39-1.21%, respectively, with an average C/N ratio of 12.1 ± 1.6%, which may be attributable to soil microorganisms present in the cryoconite. Typically, micro-organisms mineralise N from the organic matrix to support plant uptake of the nutrient. This is because soil micro-organisms require a cellular C/N ratio of about 8 which is maintained via the N mineralisation. Material in cryoconite holes in Antarctica was found to have a relatively low carbon content 0.06-0.35%, contributing to C/N ratios in the range 3.5-8.2 (Bagshaw et al., 2013), while samples from the Morteratsch and Forni glaciers had mass ratios of 9.4 ± 1.4% and 7.2 ± 0.8% for organic matter respectively, and 0.5 ± 0.25% and 0.2 ± 0.2% for elemental carbon (Baccolo et al., 2020a). Thus, there are considerable geographical differences in the C and N contents of particulate matter found in cryoconite holes both in the northern and southern hemispheres.

The presence of the cosmogenic radionuclide $^{7}$Be in these samples also provides insight into the process of accumulation of radionuclides in cryoconite. Despite being one of the most stable beryllium radioisotopes, the half-life of $^{7}$Be is relatively short at 53 days (c.f. 1.39 million years for $^{10}$Be), yet it is present in all but one of the cryoconite samples, with a mean activity of 1014 ± 599 Bq kg$^{-1}$. This is at least an order of magnitude higher than the activities of $^{7}$Be typically observed in surface soils and fine river sediments in the mid latitudes (e.g. Smith et al., 2014; Ryken et al., 2016). $^{7}$Be demonstrates rapid sorption to sediment particles and has been shown to have an affinity for reducible (e.g. Fe/Mn oxides) and oxidisable (e.g. organic) fractions (Taylor et al., 2012). Finding high activities of $^{7}$Be in cryoconite implies a recent accumulation history and supports the role of meltwater in providing a crucial link between the radionuclides stored in glacier ice and cryoconite (Baccolo et al., 2020b). The atmospheric deposition of $^{7}$Be is affected by a number of factors, including its availability in surface air for scavenging by precipitation (Aldahan et al., 2001). Concentrations of $^{7}$Be are generally higher in mid-latitude surface air masses (Kulan et al., 2006), with atmospheric circulation driving the downward transport of $^{7}$Be-rich air from the upper troposphere (Aldahan et al., 2001). This type of vertical transport is less likely in the polar regions owing to the stability of the air mass, thus, surface air of polar origin is typically expected to have relatively low $^{7}$Be activity. It is generally accepted that $^{7}$Be is largely transported to the Arctic from the mid latitudes, with a strong seasonal variation (higher in late winter/spring) that can correspond with transport of contaminants (Feely et al.,1989). Potential $^{7}$Be-rich air masses in late winter/spring with corresponding deposition, coupled with summer meltwater production and the concentrating effect of radionuclide exchange at the water-sediment interface described above, may help to explain the relatively high activities found in these cryoconite samples from Arctic Sweden. In this regard $^{7}$Be may be a useful proxy for transfer of contaminants to the cryosphere in the context of changing dynamics of atmospheric circulation (Terzi et al., 2020).

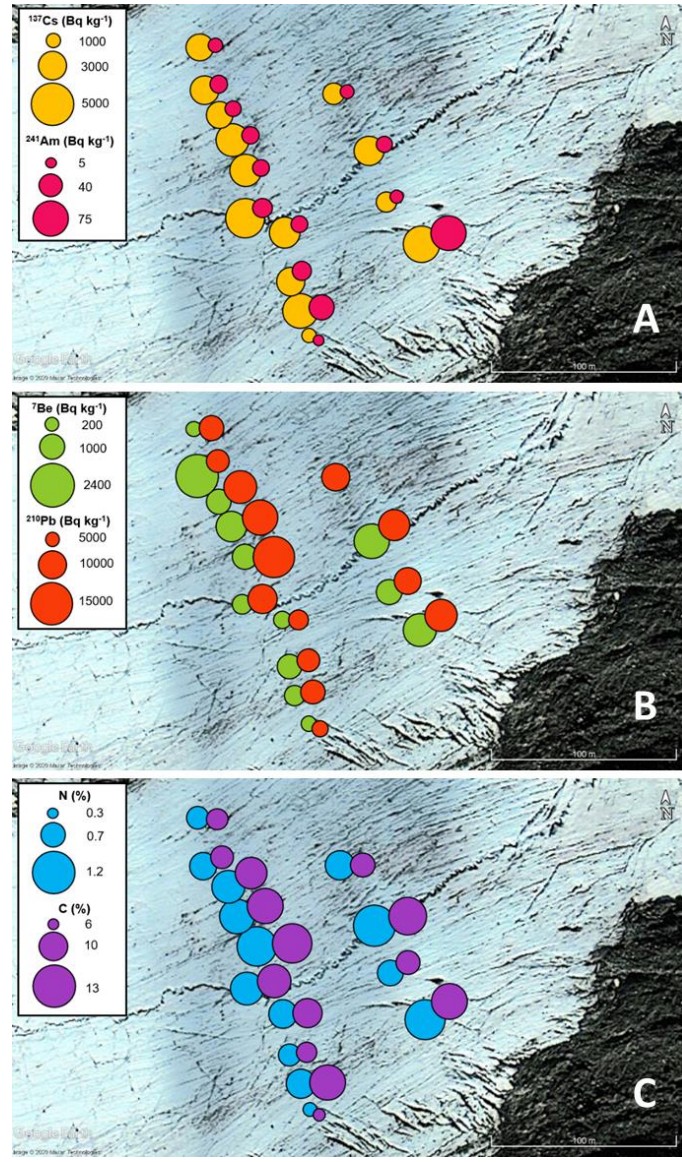

**Figure 2: Spatial variability of selected anthropogenic (A) and natural (B) radionuclide activities, and the C and N content (C) of cryoconite on Isfallsglaciären. The sampling points for cryoconite samples G1 to G14 (c.f. Fig. 1C) are centred beneath the yellow, green, and blue circles in panels A, B, and C respectively. Note that $^7$Be was not detected above the minimum detectable amount in sample G14 (panel B). Imagery source: © Google Earth, 67°54'55.25"N; 18°35'32.13"E (image from 8/10/2013).**

The average inorganic composition of cryoconite based on XRF analysis is shown in Figure 3. Not unexpectedly, $SiO_2$ is by far the most abundant element in the samples, averaging ~357900 ppm, followed by $Al_2O_3$ (~111400 ppm), and $Fe_2O_3$ (~105400 ppm). Based on calculating the normalised standard deviation for each element, the element with the highest variance between the 14 cryoconite samples is Cu (0.376), while least variance between samples (0.034) is found for $Fe_2O_3$, one of the





most abundant elements found in cryoconite on Isfallsglaciären. The sum of the concentrations of major and trace element
oxides detected via XRF spectrometry for the cryoconite samples is between 63.7 and 79.5 %, and a further 6.8-13.5 % can be
attributed to C and N based on bulk stable isotope analysis.

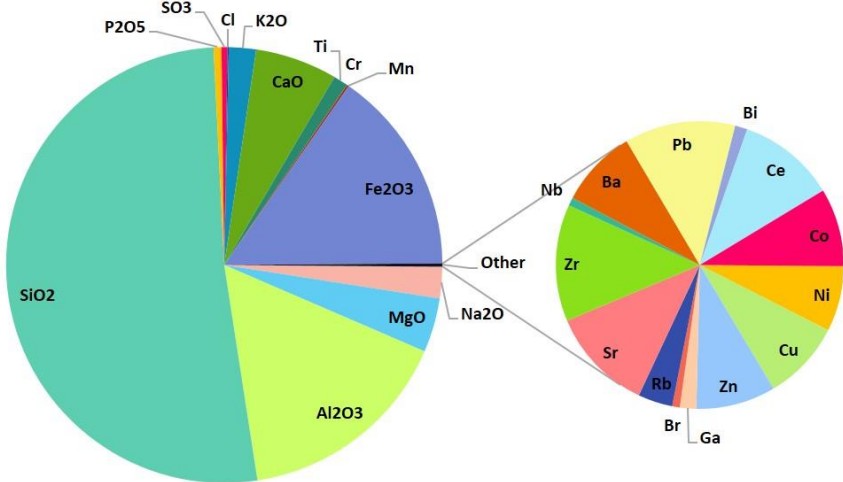

**Figure 3: Average inorganic composition proportion of cryoconite samples G1 to G14 from XRF analysis. Note that these elements
make up ~70% of the composition of each sample.**

The means of selected metal concentrations in cryoconite samples are shown in Table A3. The concentrations for Cu, Fe, Pb
and Zn are elevated over those associated with cryoconite granules found in a glacier in Svalbard (Łokas et al., 2016).  These
values together with the Al concentrations were used to determine enrichment factors *EF* from Eq. 1:

$$EF = \frac{\left\{\frac{(M)}{(Al)}\right\}}{\left\{\frac{(M)Ref}{(Al)Ref}\right\}} \qquad (1)$$

where *(M)* and *(Al)* are the total concentrations of a metal and of aluminium and *(M)*$_{Ref}$ and *(Al)*$_{Ref}$ are the reference values of
the metal and aluminium for the upper continental crust (Wedepohl, 1995). In these samples from Isfallsglaciaren the mean of
Al concentrations (n=14) was $57930 \pm 3250$ mg kg$^{-1}$. Internationally, the Canadian sediment guidelines for risk to aquatic life
can be used to evaluate whether the particulate matter at a site is contaminated or not. In general terms a sample with an EF
falling in the range 1<EF<3 has minor enrichment, the range 3<EF<5 indicates moderate enrichment, the range 5<EF<10 is
assessed as moderate to severe enrichment, and EF>10 is classed as severe enrichment. The cryoconite samples have elevated
metal concentrations but only Cr and Pb have concentrations above the probable effect level (PEL). The EF values are clustered
in three groups with the highest for Cu and Pb, followed by Cr and Ni, while Fe, Ti and Zn have the lowest EFs.





A principal component analysis (PCA) was conducted for the cryoconite samples to help explore the variance between the samples, based on gamma spectrometry, particle size analysis and stable isotope analysis. The PCA scores and loadings for principal components 1 and 2 are depicted in Figure 4, and no outliers were identified in the data based on the Mahalanobis distance. The PCA loadings (Fig. 3B; Table A4) shows that the content of C and N in cryoconite have large positive loadings on principal component 1, while area-weighted particle size (D[3,2]) has a large positive loading on principal component 2, closely followed by strong negative loadings from $\delta^{13}$C and $\delta^{15}$N. These first two components explain 70% of the variance in the data, with principal component 1 explaining 47%, and component 2 explaining a further 23%. Principal components 3, 4, and 5 explain 10%, 6.4%, and 5.9% of the variance respectively, and combined the first five principal components explain 93% of the variance in the data. Table A4 contains the eigenvector values for principal components 1 to 5, highlighting the influence of $^{7}$Be, $^{40}$K, and $^{210}$Pb on components 3 to 5. While sample numbers are limited, the PCA nevertheless reveals a clustering in cryoconite samples collected from the north side of the southern glacier terminus (Fig. 4A; c.f. Fig. 1C), suggesting that exposure to sunlight may be an influencing factor on the accumulation of FRNs, due to increased melting and/or available energy (c.f. Fig. 2). This in turn may play an important role in organic content of cryoconite. It has previously been shown that a higher proportion of bioavailable C, N, and P is present in cryoconite in comparison to source materials (Bagshaw et al., 2013), highlighting the significant role it plays within biogeochemical cycling. As this is likely due to the microbial activity within cryoconite, it is perhaps unsurprising that other elements are also found in higher concentrations where energy availability is greater.

A second PCA was performed to investigate the role of inorganic composition of cryoconite by including major and minor elements detected through XRF analysis (PCA scores and loadings for principal components 1 and 2 are depicted in Figures 4C and 4D). In this case the first three components explain 70% of the variance, and the first six 90%, with the first principal component explaining 45%, and components 2 to 6 explaining 14%, 12%, 9%, 6.6%, and 4.3% of the variance respectively. Examination of the eigenvector values for this PCA illustrate the influence of a negative loading from CaO and Na$^{2}$O for principal component 1, followed closely by MgO and Cl,, with area-weighted particle size (D[3,2]) again showing a strong negative loading for component 2. CaO, Na$^{2}$O, MgO, and Cl are all relatively soluble elements which may explain the clustering of these elements and their role in explaining variance within the samples.



Figure 4: Principal component analysis (PCA) for cryoconite samples G1-G13. Data are grouped by position on the glacier terminus (north and south; c.f. Fig. 1) for the PCA scores (A). The PCA loadings plot (B) depicts eigenvector values for FRNs (black), C and N analysis (red), and particle size analysis (blue). Principal components 1 and 2 are depicted here; see table A2 for eigenvectors for

principal components 1 to 6.

## 4.2 Catchment-wide distribution of radionuclides

The activities of radionuclides detected within the proglacial area of the Isfallsglaciären catchment are significantly lower than those in cryoconite (Table A1; Fig. 5). This supports that cryoconite is a highly efficient accumulator of radionuclides to the

extent that activities are orders of magnitude above those which were deposited and accumulated "off ice". The deglaciated central forefield has very low levels of FRNs despite having been exposed by the ice before the weapons testing era and Chernobyl; by comparison, the samples of proglacial outwash, which are fed by a regular supply of meltwater and sediment





from the glacier, are characterised by much higher activity concentrations for natural radionuclides, particularly for $^{210}$Pb and
$^{7}$Be. Indeed, of the proglacial samples, $^{7}$Be is only found in proglacial outwash and lake core sediments, suggesting that the

300 transport of sediment and radionuclides in meltwater is important for their downstream accumulation (Fig. 5) in addition to
their accumulation in cryoconite (Baccolo et al., 2020b). The importance of interaction with meltwater, and possible
enrichment by runoff of supraglacial sediments such as cryoconite, is further supported by the elevated levels of radionuclides
present in the upper portions of the proglacial lake core which are in excess of all other off-ice sediment sources sampled here.
$^{241}$Am is found only in the middle portions of the core, corresponding to known dates of nuclear activity which will be discussed

further below. These results may reflect a more continuous flux of natural radionuclides from the glacier to the proglacial area,
while FRNs, deposited during temporally-restricted events, melt out more sporadically due to their storage in defined layers
within the snow, firn, and ice. While the activity concentrations of FRNs in moraine sediments are generally low in comparison
to cryoconite and proglacial outwash, there is a clear anomaly in the radionuclide concentrations from one sample which
contained 74 Bq kg$^{-1}$ of $^{210}$Pb and 207 Bq kg$^{-1}$ of $^{137}$Cs (Fig. 5). This anomaly suggests the possible presence of localised off-

310 ice "hot spots", which may be representative of efficient accumulation of FRNs via lichens and mosses from direct atmospheric
deposition, as has been reported in other environments (e.g. Sumerling, 1984; Paatero et al., 1998; Kirchner and Daillant,
2002).



**Figure 5: Natural and anthropogenic radionuclide activities detected in sediment samples from proglacial outwash, the lake core (split into four subsections), moraines, and the central forefield, depicting the median, interquartile range, range, and outliers for each sample class. Note that the activity values for cryoconite are excluded here as they dwarf the values of the other samples (see Table A1), and an extreme outlier for 137Cs has been removed (\*moraine sample; 207.2 Bq kg$^{-1}$).**

The spatial distribution of radionuclide activity concentrations in the proglacial area of Isfallsglaciären is shown in Figure 6. $^7$Be is only present in proglacial outwash, however there is no clear pattern in which samples this has been detected (Fig. 6a). There is little variation in $^{137}$Cs between moraine, forefield, and proglacial outwash sediments, however the anomalous value described above is clearly visible in the moraine sample closest to the northern glacier terminus. There is much more obvious variation in activity concentrations for both $^{210}$Pb and $^{40}$K, which illustrate a clear difference between the sediments transported in the northern and southern proglacial outlet streams (Fig. 6b). $^{40}$K is, for the most part, present in higher levels in moraine sediments (particularly the northern moraine) than the proglacial outwash, in addition to the central forefield sample furthest





from the present day terminus (and most isolated from the braided stream system). $^{40}K$ is a common element in the Earth's crust, and the relative stability of moraines in comparison to proglacial outwash may allow for increased accumulation of this radionuclide, which has a very long half-life of 1.251 billion years. Potassium is also a relatively soluble element, and thus it may be expected that activity levels would be lowest in areas influenced by a dynamic hydrological system, while levels in 330 areas isolated from water (aside from precipitation) accumulate more $^{40}K$. The spatial distribution of $^{210}Pb$ in the proglacial area of Isfallsglaciären is more complex, with values in sediments from the southern proglacial outlet stream being notably higher than those from the north, and those in moraine sediments. This may be influenced by the glacier surface topography, as the northern terminus lobe is considerably steeper and more crevassed than the southern lobe, which may restrict the ability for cryoconite to accumulate on the surface and meltwater to flow and transfer materials uninterrupted, thus leading to 335 decreased FRN enrichment of supraglacial sediments. The southern proglacial outlet may also have a higher discharge, or be more dynamic in its flow pathways, allowing for accumulation of sediment from a larger sediment source pool.

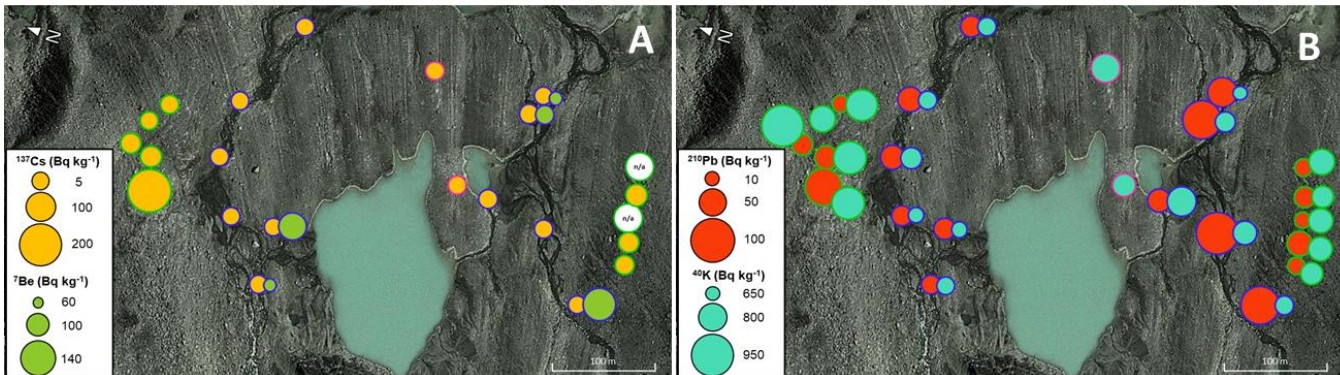

**Figure 6: Spatial variability of selected radionuclide activities detected in proglacial sediment samples from the central forefield,**
**moraines, and proglacial outwash. The sampling points are centred beneath the yellow and orange circles in panels A and B respectively. Circles with a green outline denote samples from moraines, circles with a blue outline denote samples from proglacial outwash, and circles with pink outlines denote samples from the central forefield. Note that neither $^{137}Cs$ or $^7Be$ were detected above the minimum detectable amount for two moraine samples (panel A), and that $^7Be$ was only detected in proglacial outwash samples, the lake core, and cryoconite (the latter two are not shown here). Imagery source: © Google Earth, 67°54'55.25"N; 18°35'32.13"E**
**(image from 8/10/2013).**

### 4.3 Longer-term downstream sediment and contaminant accumulation

The downcore profile of fallout $^{210}Pb$ against mass depth (reflecting accumulation rates) in lake sediments from Isfallsjön (Fig. 7a) departs from exponential decline, implying periods of enhanced sedimentation. From mass depth ca. 40 g cm$^{-2}$ (true depth
23 cm) downward, activity concentrations approached the limit of detection wherein selected samples were counted for a longer duration to achieve measurable values to model the tail for CRS modelling (not shown). The $^{137}Cs$ profile (Fig.7b) shows a first detectable activity concentration at mass depth 62 g cm$^{-2}$ (true depth 35 cm). Following Lindblom (1969), this is





inferred to represent the onset of early weapons testing in 1952. Subsequent peaks, moving upward, are inferred to represent

(i) a mid-1950s spike in atmospheric fallout (mass depth 42-45 g cm$^{-2}$), (ii) the 1963 peak in fallout (mass depth 35 g cm$^{-2}$),

which also is the first detectable activity concentration of $^{241}$Am linked predominantly to global fallout (Bunzl et al., 1995),

and finally (iii), at mass depth ca 10-12 g cm$^{-2}$, increased $^{137}$Cs activity concentration associated with fallout from the

Chernobyl nuclear accident (Olszewski et al., 2018). Within this well-constrained geochronological framework (Fig. 7c), it

can be seen that sedimentation rates (Fig. 7d) were significantly reduced c.50 years ago, reflecting the post-1959 formation of

Frontsjön following terminus retreat (Karlén, 1973), and subsequent "piracy" of proglacial waters from Isfallsjön. A critical

question remains about deposition and release of FRNs from the glacial ice during seasonal melt and more recently accelerating

retreat due to global warming. The sedimentary data suggest a degree of lagged release during the 1960s with a protracted

detection of $^{241}$Am (0.4 to 0.5 Bq kg$^{-1}$) in sediment from mass depth 33 to 26 g cm$^{-2}$ representing the period 1963 to 1970. This

might relate to release by ice or erosion of surficial sediment from the fore field. $^{241}$Am activity concentration was below

detectable limits (< ca 1.5 Bq kg$^{-1}$) after 1970 implying that any FRN activity associated with cryoconite has been diluted by

other sediment sources during melt, release and transportation, despite the limited distance between the lake coring site and

the glacier.

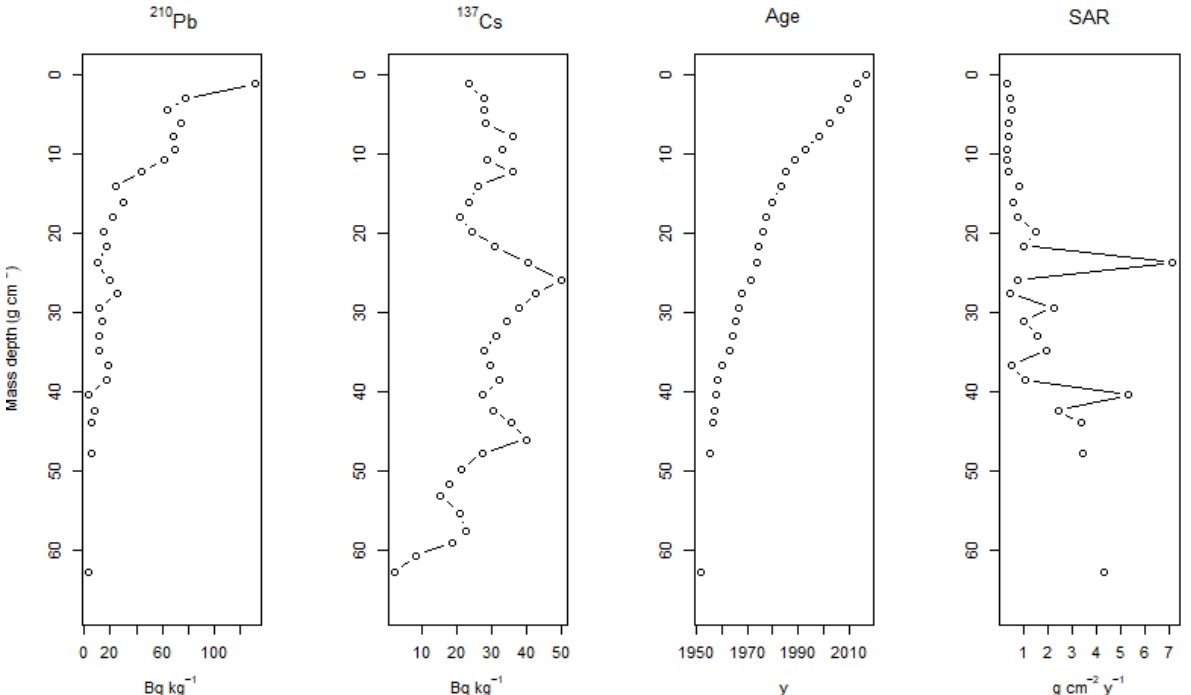

**Figure 7: Fallout radionuclide depth profiles for (a) $^{210}$Pb$_{un}$ and (b) $^{137}$Cs with and geochronological information (c) age-depth and**

**(d) sediment accumulation rate according to output from the Constant Rate of Supply model (see text for details).**



### 4.4 Implications for downstream environmental quality

There has been a considerable research effort in understanding the uptake of FRNs within flora and fauna, particularly following the 1986 Chernobyl accident. Mosses, lichens, and fungi are environmental matrices known to efficiently accumulate FRNs (Heinrich, 1992; Steinnes and Njåstad, 1993), and play a crucial role in radionuclide uptake into the food chain, particularly for reindeer and other ruminants (MacDonald et al., 2007). The effect of deposition of $^{137}$Cs from atmospheric transport following the Chernobyl accident was considerable for reindeer herding in Sweden, Norway, and other northern countries, due to contamination of the lichen-reindeer-human food chain (Skuterud et al., 2016), and a number of studies have demonstrated the importance of origin (food source) for $^{137}$Cs transfer, highlighting the lichen content of diets as a key control on uptake (Ahman et al., 2001; Skuterud et al., 2004). Despite an initially rapid drop in $^{137}$Cs in reindeer tissue in Sweden post-Chernobyl (Ahman and Ahman, 1994), concerns around the long-term impacts to exposure to fallout remain. A recent testing campaign by the Swedish radiation Safety Authority revealed levels of $^{137}$Cs of up to 39706 Bq kg$^{-1}$ in wild boar in 2017-2018, with as many as 30% of 229 boar tested exceeding the Swedish limit for meat consumption of 1500 Bq kg$^{-1}$ (Strålsäkerhetsmyndigheten, 2020). The uptake of $^{137}$Cs by lichens and fungi is likely contributing to this persistence of high levels of radioactivity in boar, in addition to the migration of boar into regions affected by Chernobyl. It has been suggested that areas with previous $^{137}$Cs contamination may augment $^{137}$Cs transfer following future contamination events due to fixation in soils (Ahman et al., 2001). The presence of radionuclides in proglacial sediments in response to ongoing glacial retreat and down-wasting could thus pose an emerging threat for ecosystem health, with the possibility of a knock-on socio-economic impact due to the health considerations of animal-human transfer, and stringent controls on limits for sale of produce for human consumption (Kristersson et al., 2017).

The levels of $^{137}$Cs detected in cryoconite on Isfallsglaciären are exceptionally high. Indeed, to the best of our knowledge they are some of the highest activity concentrations found in natural environmental matrices outside of nuclear exclusion zones. In light of our findings, and the similarly elevated levels of FRNs detected in cryoconite in other regions of the cryosphere, we recommend an increased research focus on this poorly understood contributor to contamination in proglacial environments, particularly in light of a continued trend of glacier mass loss and meltwater production. We identify a need to establish the prevalence of FRNs in glacial sediments across a wider spatial range, with a particular focus on regions where glacial meltwater is crucial to downstream water and food security, including the Andes and Himalaya. To more fully understand any potential impact of secondary FRN contamination with glacier retreat, the total mass of cryoconite in glacier catchments must also be considered when assessing whether FRNs are likely to pose any threat to downstream ecosystems. Distribution of hotspots in proglacial environments, and the concentrations of FRNs in downstream sediment sinks, must also be better constrained in order to evaluate implications for both aquatic and terrestrial fauna. Furthermore, we recommend that the bioavailability of FRNs in glacial sediments, including cryoconite, is assessed to understand whether the presence of FRNs in these settings can be taken up in the food chain to levels which are potentially harmful to fauna or for human consumption, or whether downstream dilution and distribution render these harmless. While the risk to distal communities is almost certainly low due





to the dilution in rivers, fragile, pioneering ecosystems in newly-exposed proglacial zones are more likely to accumulate radionuclides from meltwater and cryoconite transfer, and may be candidates for monitoring to evaluate risk to local fauna.

## 5 Conclusions

A holistic view of the distribution of radionuclide activities within a glacier catchment, including both the supraglacial and proglacial domains, is described here for the first time. Our study supports that FRNs are accumulated through their interaction with snow, meltwater and cryoconite, resulting in activity levels in the supraglacial environment that are up to two orders of magnitude above those found in the proglacial area. The study also highlights the widespread legacy of nuclear incidents in some of the Earth's most remote environments. It further sheds light on the influence of authigenic organic matter on radionuclide capture from meltwater in-situ in cryoconite, while the presence of [7]Be suggests recent accumulation of radionuclides in cryoconite through interaction with a regular supply of meltwater transporting legacy contaminants melting out of snow and ice up-glacier. In addition to describing levels of FRN activity in the supraglacial and proglacial environments, geochronological analysis of downstream sedimentary archives illustrates the melt and sedimentation history of the Isfallsglaciären catchment. The application of nuclear techniques to proglacial lake core chronologies can both provide insight into temporal variability in historical deposition and transport of FRNs, and how proglacial sediment accumulation has changed in response to both glacier retreat and a changing flux of meltwater production, both of which have important implications for mitigating downstream impacts of climate change.

Continued glacier retreat will result in further transport of FRNs into the downstream environment through meltwater and sediment flow pathways, but potentially also through direct deposition in the proglacial area under conditions of glacier down-wasting. Such secondary contamination events, resulting from the release of legacy contaminants stored in snow, ice, and cryoconite, may compound the issue of elevated FRN levels found in other environmental matrices such as lichens, mosses, and fungi, which are common in recently deglaciated terrain and known to impact the fauna for whom these are a key food source. This research highlights a critical need to evaluate not only the activity levels of FRNs in the supraglacial and proglacial environments, but also their total mass and spatial distribution, and whether FRNs in the proglacial environment are taken up in the food chain in quantities that are potentially harmful. This may, or may not, present an emerging environmental threat to terrestrial and aquatic ecosystems downstream of glaciers. To address this, we recommend an interdisciplinary approach to future research in this field to assess not only the distribution and variability in FRN levels in glaciers, but also the socio-environmental impact of changing quality of glacier-fed waters. In the case of Isfallsglaciären and the wider Kebnekaise area, a priority emerging from our work is to evaluate the potential impacts on FRN uptake in proglacial vegetation, and on grazing fauna such as reindeer, and the wider impact, if any, upon local Sami economy and culture. A continued effort is required to further evaluate the prevalence and spatial variation of both FRNs and other contaminants across the global cryosphere, and to better understand both the processes of contaminant accumulation in the supraglacial environment, and the downstream impacts of secondary contaminant release.



## Appendix A: Tables referred to in text

**Table A1: Comparison of radionuclide activity concentrations in IAEA worldwide proficiency tests with those determined in the laboratory used for analysis in this study (n=3). The measured values are all within the acceptable IAEA statistical criteria.**

| Radionuclide | IAEA Activity Concentration, Bq kg$^{-1}$ | Measured Activity Concentration, Bq kg$^{-1}$ |
|---|---|---|
| ***IAEA Moss Soil (IAEA-CU-2009-03)*** | | |
| $^{210}$Pb | $424 \pm 20$ | $457 \pm 12$ |
| $^{214}$Pb | $26.0 \pm 2.0$ | $22.8 \pm 1.2$ |
| $^{226}$Ra | $25.1 \pm 2.0$ | $22.8 \pm 1.2$ |
| ***IAEA Soil (IAEA-TEL-2012-03)*** | | |
| $^{137}$Cs | $118.6 \pm 2.9$ | $119 \pm 3$ |
| $^{210}$Pb | $573 \pm 25$ | $642 \pm 14$ |
| $^{241}$Am | $1.78 \pm 0.1$ | $1.97 \pm 0.37$ |



**Table A2: Activity values of selected radionuclides for sediment sources within the Isfallsglaciären catchment (Bq kg⁻¹). Values in parentheses represent the lowest recorded value above MDA (Minimum Detectable Activity).**

| | Cryoconite (n = 14) | Moraines (n = 10) | Proglacial outwash (n = 11) | Central forefield (n = 2) | Lake core (n = 38) |
|---|---|---|---|---|---|
| *Mean $^{210}Pb_{un}$* | 9777 ± 780 | 26.0 ± 11.3 | 44.5 ± 10.8 | - | 23.3 ± 6.3 |
| *Maximum* | 14663 ± 1167 | 73.5 ± 12.9 | 87.8 ± 16.2 | MDA | 125.3 ± 24.1 |
| *Minimum* | 5760 ± 467 | MDA (<10.2) | 20.3 ± 8.0 | MDA | MDA (<1.5) |
| | | | | | |
| *Mean $^{137}Cs$* | 3069 ± 940 | 36.4 ± 66.3 | 10.2 ± 4.0 | - | 25.9 ± 11.9 |
| *Maximum* | 4533 ± 350 | 207.2 ± 16.4 | 18.6 ± 2.1 | 9.11 ± 1.4 | 50.0 ± 4.9 |
| *Minimum* | 1011 ± 79 | MDA (<7.0) | 6.4 ± 1.3 | 8.36 ± 1.2 | MDA (<2.2) |
| | | | | | |
| *Mean $^{241}Am$* | 25.8 ± 16.7 | - | - | - | 0.06 ± 0.16 |
| *Maximum* | 74.0 ± 10.2 | MDA | MDA | MDA | 0.52 ± 0.24 |
| *Minimum* | 6.10 ± 3.2 | MDA | MDA | MDA | MDA (<0.4) |
| | | | | | |
| *Mean $^{40}K$* | 1839 ± 168 | 800 ± 67 | 708 ± 47 | - | 807 ± 65 |
| *Maximum* | 2054 ± 207 | 943 ± 89 | 816 ± 80 | 807 ± 71 | 928 ± 89 |
| *Minimum* | 1544 ± 138 | 728 ± 69 | 649 ± 59 | 748 ± 68 | 560 ± 63 |
| | | | | | |
| *Mean $^{7}Be$* | 1014 ± 599 | MDA (<46.5) | 94.3 ± 45.5* | - | - |
| *Maximum* | 2286 ± 307 | MDA | 132 ± 53 | MDA | 46.4 ± 13.3* |
| *Minimum* | 273 ± 58 | MDA | MDA (<67.2) | MDA | MDA |

*Only n=5 samples gave a $^{7}Be$ signal; **$^{7}Be$ was only recorded (above MDA) in the top 1cm section of the lake core.*



**Table A3: Mean (±1 std dev) concentrations of selected metals (mg kg⁻¹) in cryoconite samples (n=14) from the Isfallsglaciären and their enrichment factors. The sample concentrations are compared with the probable effect level (PEL) from the sediment quality guidelines (Hübner et al., 2009).**

|  | Cr | Cu | Fe | Ni | Pb | Ti | Zn |
|---|---|---|---|---|---|---|---|
| *Concentration, mg kg⁻¹* | 212±27 | 155±63 | 73700±2650 | 127±15 | 215±51 | 7440±350 | 155±20 |
| *Enrichment Factors (EF)* | 8.1±0.9 | 14.4±5.0 | 3.2±0.2 | 9.2±1.3 | 17.1±4.4 | 3.2±0.1 | 4.0±0.5 |
| *SQG PEL mg kg⁻¹* | 160 | 108 | 0 | 42.8 | 112 | - | 271 |

**Table A4: Eigenvectors for principal components 1 to 5 from PCA of cryoconite data including gamma spectrometry ($^{137}$Cs, $^{210}$Pb, $^{241}$Am, $^{40}$K, $^{7}$Be), stable isotope analysis (%N, %C, $\delta^{13}$C, $\delta^{15}$N, and surface area-weighted particle size (D[3,2]). The variables that have the largest effect on each principal component are highlighted.**

|  | PC1 | PC2 | PC3 | PC4 | PC5 |
|---|---|---|---|---|---|
| *% N* | 0.409 | -0.142 | 0.277 | -0.25 | 0.003 |
| *% C* | 0.431 | -0.112 | -0.001 | -0.33 | 0.015 |
| *$^{137}$Cs* | 0.367 | 0.076 | 0.397 | 0.107 | 0.475 |
| *$^{210}$Pb* | 0.36 | -0.224 | 0.079 | -0.326 | -0.494 |
| *$^{241}$Am* | 0.323 | 0.405 | 0.04 | 0.339 | 0.046 |
| *$^{40}$K* | 0.312 | -0.173 | 0.072 | 0.746 | -0.372 |
| *$^{7}$Be* | 0.241 | -0.116 | -0.731 | 0.062 | 0.169 |
| *D[3,2]* | 0.182 | 0.516 | -0.319 | -0.136 | -0.411 |
| *$\delta^{13}$C* | -0.227 | -0.472 | 0.103 | 0.106 | -0.312 |
| *$\delta^{15}$N* | 0.196 | -0.464 | -0.327 | 0.073 | 0.308 |





**Data availability**

Gamma spectrometry, XRF, stable isotope, and particle size analysis data will all be available via Pangaea (currently waiting for a DOI to be assigned). Data are also available by request to CCC.

**Author contributions**

CCC, WHB, and SCM devised the study, CCC and NS conducted field sampling, CCC, GEM, AT, and JK conducted preparation and analysis of the samples, PB designed the organic analytical work programme, CCC and WHB led interpretation of the data, and all authors contributed to preparation of the manuscript.

**Competing interests**

The authors declare that they have no conflict of interest.

**Acknowledgements**

We thank Richard Hartley at the University of Plymouth for his help with particle size analysis, and Katja Van Nieuland at the Ghent University for conducting bulk stable isotope analysis. Sample collection was funded by an INTERACT Transnational Access grant awarded to CCC, and we thank the staff of Tarfala Research Station for providing logistical support during our fieldwork.

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
