# Peer review of "Accumulation of legacy fallout radionuclides in cryoconite on Isfallsglaciären (Arctic Sweden) and their downstream spatial distribution"

_The Cryosphere, 2021_

## Author Comment (AC1)

We thank Anonymous Referee #1 for their considered, detailed, and helpful review of our manuscript. Below we outline our response to each comment in turn (shown in blue text), with the reviewer's original comment in italics.

*GENERAL COMMENTS*

*This manuscript provides an original dataset including gamma-emitting fallout radionuclides activities, geochemical oxide composition/trace element concentrations, and C/N compositions obtained on cryoconite samples (n=14), sediment sources (n=23) and a 38-cm long lacustrine sediment core collected in a 1-km² glaciated catchment of Sweden. The authors discuss the spatial and temporal distributions of these properties in response to glacial processes and sediment transport. They also underline the potential environmental and health deleterious impacts of the redistribution of these potentially harmful substances (e.g. in response to the ongoing global warming and snowmelt processes). Overall, the manuscript is well written, the data is well described and the results are well discussed at the light of those previously published in the literature. However, in my opinion, the study site should be further contextualized. Furthermore, the maps could be improved and the interpretation of some of the results could be taken further (e.g. relying on part of the data analysed although maybe under-exploited, regarding the particle size and the organic matter composition, in particular for cryoconite samples). The interest of this research for tracing the impacts of snowmelt-induced sediment redistribution could also be underlined (instead of focusing only on the potential harmful impacts of the release of radionuclide substances stored in this glaciated area in response to snowmelt, although I acknowledge this interest of doing so, in particular in Sweden, for the reasons well explained by the authors in the manuscript including the substantial post-Chernobyl fallout in this region and the specificities of the regional food diet). Accordingly, I recommend reconsidering acceptance of the manuscript after major revisions have been performed. Nevertheless, I would like to underline the fact that I enjoyed reading the manuscript, and improving the data presentation /interpretation will definitely further increase the impact of this nicely conducted study. I acknowledge the large quantity of fieldwork (in very likely adverse conditions!) and labwork conducted, and I am thankful of having been given the opportunity to discuss these results with the community!*

- We respond to the general comments made here within our response to individual comments below.

**DETAILED COMMENTS**

*Abstract*

*LL18-20 maybe adding values would be useful for the readers here; not sure whether you only refer to the Chernobyl and Fukushima exclusion zones here?*

- We don't feel that adding specific values is necessary within the abstract, rather focussing on the key take-home messages of the work.

*LL.25-26: as in the general comment above, I would also insist on the interest of analysing these substances as unique tracers to understand the impact of ongoing snowmelt processes on the dispersion of sediment and associated contaminants across catchments in this part of the world.*

- We feel that discussion of FRNs as tracers of snowmelt processes and sediment distribution is outside the scope of this manuscript, despite being a very interesting topic. Furthermore, samples

were collected at the end of the ablation period when much snow cover had already disappeared, and no samples were collected within the glacier accumulation zone, thus we don't feel we can assess this aspect further. We acknowledge that radionuclides and FRNs specifically can be used in a wide range of environmental applications, but here we focus on FRNs as contaminants. We direct the referee and other interested readers to a recent paper that specifically focusses on application of [210]Pb to different areas of glaciological research: https://doi.org/10.1017/jog.2020.19

**Introduction**

*L.42 not sure I would start this sentence with a 'but'?*

- Changed to "However".

*L.50 could you provide examples of what would be 'anthropogenic organics'?*

- The term "anthropogenic organics" has been replaced with "persistent organic chemicals" and the specific compounds, well known to the scientific community, can be found in the references, as is also the case with the metals.

*59 please remove the . before the (*

- Removed.

*L.60 (and elsewhere in the text): Is 'prevalence' the right term to use here? Wouldn't the terms 'persistence' or simply 'occurrence' be more appropriate?*

- Text updated, apart form in the conclusion where we feel that 'prevalence' is the correct term.

*L.62 'environmental fate of the radioactivity' >> fate of the radionuclides?*

- Personal preference; we prefer the text as currently written.

*63 unclear what you mean with 'in addition to any potential socio-economic impacts' here? Could you be more specific?*

- Our paper does not try to explore what the socio-economics impacts would be, only to highlight that these types of impacts are possible. Thus, we don't feel it would be appropriate to suggest specific impacts at this stage, however we have added general comments to this matter within the text.

*LL.64-65 same remark with 'subsequent dilution in the hydrological system', could you specify what you mean?*

- We have now made it clear in the text that this dilution relates to when cryoconite enters proglacial waters.

*LL.67-76; I had the impression that this paragraph could be rewritten to better outline scientific questions instead of listing the analyses made and 'arguing' that FRN activities are higher than those found in other environments?*
- The statement "we argue" has been removed in line with a suggestion from referee 2, and the text changed to reflect that this finding is a combination of work from different studies.

***Study site***

*In my opinion, this section was pretty short and focused on glacial geomorphology. What about rainfall/snowfall in this catchment? What about the bedrock lithology, the soil characteristics (if this is relevant in this catchment?), etc.*

- We agree that the addition of information on climate and geology of the catchment would be helpful here, and have added this to the manuscript.

***Methods***

*Overall, in this section, the sampling period should be better defined (in the introduction, the period of 'August 2017' is mentioned, then nothing else is added… How high was the precipitation during the months before sampling? How was it distributed with time? I guess that this is a crucial aspect for supporting the interpretation of the results (e.g. mainly for Be-7 and Pb-210 data).*

- More information on the sampling period has been added to the opening paragraph of the methods section. We do not have access to weather station data local to the site to further analyse the influence of precipitation on $^7$Be and $^{210}$Pb activities, and would only be speculating on the content of $^7$Be and $^{210}$Pb in precipitation.

*L.98 the authors refer to 'sources' here, although the reference to sources is no longer used in the interpretation… Were all the potential sources covered by this sampling (or maybe this wasn't the purpose…)?*

- We list the sources considered here in the sentence immediately following this statement. A potential source not included is rockfall from surrounding slopes, however this cannot be sampled safely at this location.

*L.102 could you be more specific on what you mean when referring to the 'proglacial outlets'?*

- Replaced with "proglacial outlet streams".

*L.106 At 100°C? Isn't it too high for the organic matter composition analyses?*

Had the particulate samples remained moist their composition may have been affected during their transport to the UK. Thus, the samples were carefully dried, immediately after collection, on site, to remove the interstitial water. Water removal is essential prior stable isotope analysis and the samples were sent for analysis directly on return to the lab

*L.107 'due to the limited amount of cryoconite available' >> what did it represent in grams of material (to have an idea of the difficulties encountered)?*

- We can't put a number on this; it more reflects the difficulty of sampling cryoconite when it is smeared on a rough ice surface, or easily displaced when in water etc. Cryoconite was typically not found in large individual accumulations at this site. However, cryoconite is typically found on glaciers as an aggregate consisting of a few grams each. Sometimes they are easy to sample because they are concentrated (cryoconite holes), but sometimes (as in this case) cryoconite is spread on the surface of the glacier requiring careful consideration during collection.

*L.107 why using a < 75μm sieving threshold (compared to the classical 63 μm threshold for instance?)*

- It would have possibly been better to apply a standard 63 μm sieve, but only a 75 μm mesh sized sieve was available on site.

*L.109 was there a specific preparation protocol implemented before conducting the particle size analyses?*

- Samples were prepared as previously outlined in the methods in terms of drying and sieving, and analysed in triplicate as stated within the manuscript.

*L.115 the article cited here (Wynants et al., 2020) refers to a study conducted in the African Rift Region with limited additional details on the gamma spectrometry analyses: I am not sure this has an added value here, or did I miss something?*

- This reference is included with respect to analytical method only (as it was also conducted using the facilities at the University of Plymouth), and not because this paper is of relevance to glacier systems.

*L.130 I guess that you used two certified material samples to account for the matrix composition differences between cryoconites (more similar to moss soil?) and other sediment materials (more similar to the 'soil' TEL-2012-03); is it so?*

- The CORiF laboratory regularly participates in IAEA international proficiency tests using their certified materials. The purpose of the repeated analyses of two IAEA soils in this work was to confirm our analytical quality assurance, thereby giving confidence that the gamma counters used in the analysis of key radionuclides were performing at the required standard throughout the analytical phase of the study.

*LL.134-147: the section on WD-XRF analyses is well described, except maybe the calibration/validation issue: was it exclusively based on comparing the results obtained with WD-XRF and ICP-OES on a selection of samples, or were certified materials also used for this crucial step?*

- Empirical calibration for XRF spectrometry requires the use of matrix-matched standards of known composition and concentration. Given the heterogeneous nature of cryoconite it is not practical to obtain matrix-matched standards for instrument calibration in this instance. Therefore, we used the analysis package, Omnian, developed by PANalytical, which is designed to handle a range of different matrices. Measurement validation can either be obtained by using reference materials or by comparison to results obtained from a different technique. Given the importance of sample composition in XRF analysis, reference materials would need to be matrix matched and there are no suitable reference materials available for this. We, therefore, validated our measurements by comparing them to results obtained from ICP-OES analysis. ICP-OES determines element concentrations in the dissolved phase following digestion of cryoconite, overcoming any matrix effects associated with analysis of the solid material. The ICP-OES procedure was itself a validated approach undertaken in an ISO9001-2015 certified laboratory. Close agreement between results obtained from the two approaches provides confidence in the XRF data.

*L.149 wasn't the drying of cryoconites at 100°C a problem for the subsequent stable isotope analyses? Where are the analysed d13C /d15N values of the samples provided? Or did I misunderstand this paragraph and only the TOC/TN were analysed?*

- Stable isotope data are publicly available on the Pangaea data repository alongside all other data from this study, and both %C/%N and d13C/d15N were included in the PCA analysis. Please see our previous response regarding drying of the samples.

*L.164: the attribution of the year 1952 or 1954 (or even 1955) may be debated for the lower level of lacustrine sediment in which Cs-137 is detected. This gives at least an idea of the uncertainties associated with core dating (+/- 4 years?). What about the level to which the year 1959 (L.169) is attributed? To which event is it related (e.g. the Tsar bomb fallout took place in 1961)? Or did I miss something in the rationale here?*

- These are important points of detail that raise issues of clarity in our message. The initial mention of 1959 in the text refers to the known geomorphological event of stream capture by the new lake and hence the rationale for the split approach to dating. The second reference in this paragraph to 1959 was more of a 'wiggle matching' exercise to fallout records in cited literature but not presented. While the reviewer offers interesting detail on the potential cause of this small peak, on reflection of the above comments we think this is a distraction from the main geochronological messages so we have modified the text accordingly to focus on the known date of onset (notwithstanding potential uncertainty).

***Results and discussion***

*L.177 is the title inclusive enough here ('geochemical composition of cryoconite') given the section contents?*

- Changed to "Cryoconite composition".

*LL.183-184/L.185/etc. at some places, some generalities are given on radionuclides, although I wonder whether they are really relevant (e.g. on the solubility of Cs-137 and the long range transport of Chernobyl fallout…?)*

- We feel that this information is relevant for context as the general readership of the Cryosphere may not have prior knowledge of FRN sources and impacts.

*LL.193-195 Providing additional information on the lithologies found in the catchment would be very useful to support the interpretations made on the K-40 levels measured in the samples*

- This information has been added to the site description in line with the previous comment.

*LL.199-200 maybe I missed it here, but I don't see how Fig. 2C illustrates the influence of organic content on the accumulation of radionuclides; additional elements should be provided here to better support this statement…*

- We agree that this statement is not helpful here and have edited the text.

*L.203 not sure it is relevant to use 2 decimal digits here for the %C and %N?*

- We have changed this to one decimal digit.

*L.212-230: to better support the interpretation of Be-7 results here, additional information should be provided on precipitation (snow/rainfall + snowmelt) during the months before the sampling campaign was conducted here. Furthermore, given the short half-life of Be-7 (~53 days), information on the analysis period after sampling is also crucial (as Be-7 could be <MDA in some samples just*

*because some samples were analysed too late, and Be-7 initially present had just decayed to undetectable levels?)*

- We agree that comparison with precipitation data could be useful here, however precipitation data are not available via the Tarfala Research Station database post-2013 currently, and we did not collect this data ourselves. We also believe that the $^7$Be detected in cryoconite samples is likely sourced from recent snowmelt, which precipitation data alone would not help to explore. We are aware, due to considerable team experience of working with 7Be, that samples must be prepared for gamma counting shortly after collection. Thus, prior to the fieldwork (7 - 17/08/2017) it was arranged with the analysts that certain samples would be packed and analysed as soon as practically possible after return to Plymouth.

*L.220 maybe the following ESSD manuscript (on Be-7 and Pb-210 levels across the globe) could be of interest to the authors here: https://essd.copernicus.org/preprints/essd-2021-35/, similar work must have been published on Cs-137 as well.*

- We thank the reviewer for drawing this publication to our attention.

*L.248 how were these metals selected?*

- The narrow range of metals is reported in Table A3 arises because there is only a limited number of metals (Cr, Cu, Ni, Pb and Zn) that have appropriate sediment quality guidelines (CCME, 1995). We felt that it was important to set the results of the cryoconite analyses within the context of probable effect levels (PELs) thereby indicating whether a toxic effect would be exerted on local fauna.

*L.253 is the normalization to the upper continental crust the most relevant option here to calculate the enrichment factors? Or did you use a more specific dataset for normalization?*

- We used the Wedepohl (1995) values for the upper continental crust because this data has international relevance.

*L.275 'it is perhaps unsurprising' > I would consider rephrasing the sentence here?*

- Now reworded in the text.

*L.281/283 problem with the notation of Na2O here (should be a subscript instead of a superscript?)*

- Both instances now fixed.

**Discussion in subsection 4.2**

*LL.293-… Overall, here, I think that the comparison of properties in cryoconite vs. other samples should be supported by the comparison of their respective particle size/organic matter compositions, as an enrichment in fine and organic material in cryoconites vs. other samples is expected to control the higher levels of contaminants measured in these samples and, according to the Materials and Methods section, you did analyse these properties…*

- We have now added a correlation matrix for each sample type as an appendix to the manuscript (Figure B1). Organic matter composition is only available for the cryoconite samples, so no comparisons can be made here, however the correlation matrices demonstrate where relationships exist with particle size (D[3,2]). Text has also been added to section 4.2 to further discuss this.

*LL.299-301 see my remark above on the precipitation before sampling to support the interpretation of the Be-7 results*

- Please see our previous response with regards to $^7$Be and precipitation data, however what we would say is that even if precipitation data had been collected, it would be speculation as to whether it might affect the $^7$Be loadings without understanding the radionuclide composition of precipitation.

*L.323 see my remark above about referring to the lithological characteristics of the study site to interpret K-40 (and I guess supported) Pb-210 here…?*

- Lithological information has been added in the study site description.

*LL.363-366: not sure if you can go that far in the interpretation regarding Am-241 activities here?*

- We are comfortable with this interpretation since the protracted delivery of $^{241}$Am to the lake sediment column is not the norm. It implies a continued release from the secondary source – certainly an inference and not fact but worthy of mention given the interest in delivery and transit of FRNs. We have modified the text to be clear this is an inference.

**Section 4.4**

*LL.372-… importantly, here, I would add the implications of these results and the detection of such high FRN activities in material transiting glaciated environments to trace and understand the icemelt/snowmelt-induced redistribution processes in the future?*

- We don't yet know what the implications of findings specifically in glacial environments are, so we can only discuss what is understood from other studies. Again, we feel that discussion of snowmelt processes are not within the scope of this study, especially since it was conducted during August which is well into the ablation season when much of the snowpack at lower altitudes has already melted (we did not sample anything within the accumulation zone of the glacier because cryoconite, if indeed it exists there, is hidden by remnant snow).

*LL.384-386 'It has been suggested that areas with previous 137Cs contamination may augment 137Cs transfer following future contamination events due to fixation in soils' >> unclear what you mean here, could you please clarify this statement?*

- The $^{137}$Cs soil fixation process implies a strong adsorption of the radionuclide to the solid phase. Fallout of $^{137}$Cs from future contamination events would add to the existing $^{137}$Cs burden of soils, thereby augmenting the transfer of this long-lived radionuclide to Arctic fauna. The text has been edited to clarify this point.

**Conclusions**

*Based on the changes made when revising the manuscript, some conclusions could be revised here (e.g. interpretations related to the organic matter content in cryoconites vs. other samples; interpretations related to Be-7 activities…)*

- We don't feel that any of the revisions made warrant changes to the conclusions.

*L.412: nuclear incidents >> nuclear accidents? Importantly, the thermonuclear bomb testing supplied most of these FRN (and I am not sure that they can be considered as nuclear incidents or accidents?)*

*- Text updated.*

*L.434 see the previous comment regarding the use of the 'prevalence' term*

*- See prior response.*

**Figures**

*Figure 1: this catchment map actually does not provide catchment delineation (to the best of my understanding of this map); could this be improved? Furthermore, there is no North arrow/ scale on the inset map of Sweden (I guess this is the map of Sweden?); it is hard to see the river network and the sample symbols are not so easy to see/understand on these maps... This is nice to see the glacier lobes on the image but the other features of interest (sampling locations and types, catchment delineation, river network...) could be presented in a much clearer way in my opinion...*

*- This figure has been updated to improve clarity of the sample points and labels, and we have removed the inset map of Sweden to make space for a larger figure legend. The stream network is a highly mobile, braided system which changes regularly, thus there is limited value in mapping stream channels from imagery that is not from a time period that coincides with when we were in the field. We have delineated the catchment boundary and included this in the new figure.*

*Figure 2: it is not easy to read/infer the FRN activities based on this map (as single values are attributed to the circles), the exact values could maybe be added near the circles on the map in red/blue?*

*- The key thing for readers to take away from this figure is the relative variability in activities, so we don't feel that individual values are required here. All data are publicly available should anyone want to examine this in more detail.*

*Figure 5: I don't understand why the core samples were split into 4 sections; in my opinion, the core samples should be merged and compared to the potential sources (maybe the continuously supplied FRN such as Pb-210 and Be-7 as they decay with time/with depth in the core and they are therefore not fully relevant for this comparison but some of the geochemical elements/ organic matter properties might be?); I would consider adding the cryoconite properties to the graphs (after normalization to the particle size/organic matter properties?)*

*- We split the core because the depths represent different periods of time and inferred geomorphological process relevant to changing dynamics of contaminant delivery. The top section of the core may also contain materials that are more easily remobilised, for example. We do not include cryoconite FRN values here because they are between one and three orders of magnitude higher than activities in other sediments and we are comparing material from compartments of the sediment continuum.*

*Figure 6 is not very easy to read neither, at least the river network should be clearly added to the map?*

*- The river network is a dynamic, braided, proglacial system and the streams seen on this particular satellite image will not precisely reflect how it looks today, or even at the time we sampled.*

*Figure 7: figure resolution seems not to be optimal, the years attributed to the peaks could be added on the Cs-137 part of the graph?*

- We prefer to leave the dates off as there are inferences in the geochronological discussion, as per above response, and explained in the text. If the figure resolution is still not optimal when we upload the figures individually for the final manuscript then we can look into redrawing the figure.

***Appendices***

*Table A.1 what was the decay-correction date? Not sure that you should provide 2 decimal digits for the Am-241 activities?*

- The samples were decay corrected to the sampling dates between 7 - 17/08/2017. The decay correction approach was stated in the original text. The $^{241}$Am activities recorded in Table A2 have been adjusted to one decimal place, as suggested.

*Table A.2 I would remain consistent with the number of decimal digits provided in this table… Do the values after +/- refer to the SD? For the 'central forefield' samples (n=2!), this is meaningless to provide a mean/SD and I would provide the range of values instead…?*

- It was not well explained in the original caption that the +/- values refer to the counting errors. Thus, for the central forefield samples the activity concentrations are reported together with their individual counting errors. We have maintained consistency with the number of decimal digits.

*Adding the particle size/organic matter data to this summary table would be very useful…*

- The particle size and stable isotope data are publicly available via Pangaea for those who are interested to look at this in more detail, however we feel that we have illustrated the importance of organic content within both figures 2 and 4, and would prefer that table A2 focusses on radionuclides alone. We have, however, now added correlation matrices illustrating the possible influence of these variables on FRN activity as an appendix.

*Regarding the footnote on Be-7: information on the time between sampling and analysis would be particularly meaningful here…*

- An appropriate comment has now been made in the footnote to table A2.

*Table A.3 were the EF calculated based on the data of the upper continental crust here?*

- The EFs were estimated using the data for the upper continental crust (Wedepohl, 1995). The original caption has been updated to include the source of sediment quality guidelines.

---

## Author Comment (AC2)

We thank Anonymous Referee #2 for their review and helpful comments on our manuscript. Below we outline our response to each specific comment in turn (shown in blue text), with the reviewer's original comment in italics.

*The manuscript of Clason et al. deals with interesting for a wide community of scientists topic of contamination of cryospheric systems. Data presented in the manuscript are novel and important in the recognizing concentration and distribution of artificial and natural radionuclides on glaciers and glacier adjacent habitats. The authors focused not only on the nuclides but also on the geochemistry of samples. I appreciate all efforts taken in the design of the study, fieldwork, and the text. However, some parts need reconstruction, better description, and careful discussion. I recommend the manuscript for publication but only after crucial improvements.*

**GENERAL COMMENTS**

*Strong points:*
- *a proper sampling design mirroring environmental gradients,*
- *investigation of the proglacial lake sediments,*
- *first data on the artificial radionuclides content in cryoconite in Scandinavia,*
- *very nice and well-prepared figures,*
- *providing all necessary raw data in the supplementary material.*

*Weak points:*
- *some parts of the text are overstated,*
- *methods require better description,*
- *authors did not show statistical differences between sampling points and types of the material,*
- *in the central forefield, only two samples were collected, making this area weak for any comparison*
- *many statements require appropriate references.*

- We respond to the general comments made here within our response to individual comments below.

**SPECIFIC COMMENTS**

**Title**

*Sounds good, however, in the light of the recent literature about the artificial radionuclide content in the cryoconite environments, authors can not say about hyper-accumulation which is overstated. I suggest rewriting the title and say about the spatial distribution of artificial and natural radionuclides in glacial and glacier adjacent environments. It is something new.*

- The very high activity concentrations of radionuclides detected in Swedish cryoconite complement other recent studies that have also detected high accumulations and the title already mentions downstream distribution. We have, however, removed "hyper" and added "spatial" to the title.

**Abstract**

*Line 10-15: I would be happy to see facts. For example, the first sentence suggests a threat for downstream systems while the results do not really indicate such a phenomenon. Monitoring is very*

*important, and this paper indeed contributes to broadening this knowledge. I suggest rewrite some parts of the text.*

*Moreover, I feel that authors should focus on the rationale of the study, aim, a brief description of methods and results, finally conclusions based on empirical evidence.*

- We feel that the key findings of the paper are included within the abstract, and also feel that it is important to set the results within a wider context. The opening statement has been reworded and set within the context of legacy contaminants more broadly, as have the final two sentences.

**Introduction**

*This part is well written and presents a robust background for the study. I would add only a short section presenting why the concentration of radionuclides is so high in the glacial environments and why glaciers are a good study site for the investigation of FRN.*

- As alluded to at the end of the abstract, the processes governing the accumulation of FRNs in glacial environments remain poorly understood, so there are currently no published studies to reference here that explore in any detail **why** radionuclide concentrations are so high.

*Lines 60-65. Authors overlooked data from Antarctica (Buda et al. 2020, Biotope and biocenosis of cryoconite hole ecosystems on Ecology Glacier in the maritime Antarctic. Science of The Total Environment, 724, 138112.).*

- The authors thank the reviewer for alerting us to this source and the reference added to text.

*Line 75. I feel that knowledge on cryoconite as the efficient accumulator of various contaminants (artificial radionuclides, heavy metals, POPs, etc.) is widely known. In my opinion, authors shouldn't describe this fact as a part of their own findings.*

- We have modified the text to make this clear.

**Study site**

*This part requires special improvements. The authors roughly described the glacier and the surrounding area. Study on the spatial distribution of nuclides in the environment requires a much more careful description of glacier bedrock geology, amount of rain, snow cover, potential sources of contaminants (I believe the study site is great since is located between Novaya Zemlya, Chernobyl, and is far from towns and factories potentially delivered heavy metals), organic matter content. Moreover, cryospheric systems are much more simple than other Arctic systems, like for example fjords, tundra. The simplicity of biological communities, easy way to find sources of microbes and organic matter, stable temperature, and predictable behaviour of the glacier makes it a good study model.*

- While we have added some information here on local geology and climate, the snow cover will change on a regular basis so we have not added any information regarding this. There is also no previous data on organic matter content at this site so any description of this is reserved for discussion of our own results later in the manuscript. As far as heavy metals are concerned, significant atmospheric inputs from the major Russian industrial complexes on the Kola Peninsula have been reported. However, the Tarfala site is some 600 km to the west of these inputs and

relatively low metal atmospheric depositions of metals from the Kola industry have been recorded in northern Norway (Chekushin et al., 1998. Sci Total Environ., 220, 95-114).

***Methods***

*When samples were collected? how many samples have been collected? how they were stored?*

- This information has now been added to the text.

*Provide a range of a.s.l. for the sampling sites.*

- This information has now been added to the text.

*Line 108: what means „sufficient material" ? provide amount/volume/weight?*

- This varies by sample type. Some cryoconite is very "fluffy" while other samples are more granular, such that a full vial will vary in mass.

***Stable isotope analysis***

*How material was collected and stored? It was frozen? It was kept at a low temperature?*

- The material used for this analysis was the same as described in section 3.1. It was not frozen, was dried on site, and stored in individual clean plastic bags for transport to the laboratory in the UK.

*Do authors prepare any replicates in order to get the most accurate analysis of 13C/12C and 15N/14N?*

- The final sentence in section 3.4 now describes the duplicate analyses.

***Results and Discussion***

*Lines 178-79. Be more specific and add the number of collected samples in methods.*

- The number of cryoconite samples has now been stated within the methods section.

*Lines 184-185. Provide appropriate reference.*

- I'm not sure which part this refers to specifically, but we have added a reference about the impacts of [137]Cs for health (Van Oostdam et al., 1999).

*Lines 200-204. Could authors make this long sentence shorter? Split into two?*

- We have shortened this sentence as suggested.

*Line 204. Why only soil organisms? Many recently published papers indicate cryoconite hosts unique, independent from other freshwater and soil habitats microbial communities.*

- This is true, and an important component of cryoconite ecology research. We have changed the text to reflect the wider, diverse microbial community.

*Lines 209-210. Taking into account that only three glaciers were investigated so far, I feel that comparison between hemispheres is too far.*

- We agree, and have modified the text to remove mention of the hemispheres.

*Lines 228-229. I would be happy to see this idea better described.*

- The lines of text to which the reviewer refers have been re-worded to improve the clarity.

*Lines 250-252. This part should be transferred to methods.*

- We appreciate the reviewer's opinion on the position of this statement within the text, however as it explains a new level of analysis we would prefer to keep this section where it is to ease the reader's understanding. The lines immediately preceding describe the outcome of the XRF analyses, that is the sum of the concentrations of the major and trace elemental oxides and as such the information is appropriate to the Results and Discussion.

*Line 254. What is „Canadian sediment guidelines for risk to aquatic life"? please provide a reference.*

- The authors thank the reviewer for drawing attention to this omission. The CCME (1995) reference has been added to the text and the list (line 261). The reference Hűbner et al (2009) is inappropriate and has been removed.

*Lines 269-274. The effect of sunlight seems to be something new. Maybe it is worth discussing the exposure of other glaciers investigated in terms of FRN and see this idea in a wider context.*

- While we agree that this is an interesting and novel finding, we feel that this is outside of the scope of this study, and would require both individual FRN sample data and geolocations of samples from other sites. Future research by our research network will focus on comparison between sites.

*I feel that sunlight influences productivity, then higher chances for accumulation of FRNs by photoautotrophs and other microbial species. The paper of Huang et al. (Accumulation of Atmospheric Mercury in Glacier Cryoconite over Western China. Environmental Science & Technology 53(12)) will be also very helpful for discussion.*

- We thank the reviewer for drawing this paper to our attention.

*Lines 277-285. I think that authors should use more statistical analysis than only PCA. The concentration of FRNs between types of environments and material can be neatly presented.*

- The reviewer does not specifically state which further analyses they would recommend, and we do illustrate the differences in FRNs between environment / material types in figure 5. However, we have now added correlation matrices for each sediment type to aid comparison between environments.

*Line 310. It is one observation only, I suggest being careful in the explanation of this phenomenon.*

- We agree, and have updated the text to urge caution over this interpretation.

*Lines 390-405. I feel that this part is no needed. At this moment it is rather a speculation. In my opinion and according to the results of the authors, FRNs will be too diluted in downstream to be*

*harmful. Nevertheless, monitoring and control of this issue are very important. I suggest remove this part or write it in another way.*

- This section of text does not suggest that FRNs **will** be harmful, but rather states that more research is required to better understand FRN distribution and accumulation in glaciated environments. We do not believe that sufficient research has yet been conducted to confidently state that FRNs in glaciated are not harmful, thus we would like to keep this section in the paper to help foster discussion and future research ideas within the community.

**Conclusions**

*The authors wrote few sentences which are not the effects of their work. I suggest adding proper references in appropriate parts of the text.*

- We have removed text within the conclusion around FRNs across the cryosphere since we only focus on one site here. We would also prefer not to add any references to this concluding section as there is nothing new stated here that is not already covered within the manuscript.

---

## Editor Decision (ED1)

Dear Dr Clason and colleagues,

Thank you for your thorough response to the Reviewer's comments. For the next stage of review, I invite you to upload your updated manuscript.

I do find some of the responses a little defensive, and would urge you consider whether a few more of the changes requested by the reviewers can be incorporated into the text. Both reviews are constructive, and I think many of the suggestions will improve the understanding of the paper for a wider audience. For example, I ask you to consider the following:

*Reviewer 1*

L25-26 comment. Will you add the suggested paper to the text? It would be useful.

L62: why not 'fate of the radionuclides'?

Methods: please ensure the information about precipitation and the justification you have here is added to the text. It would also be useful to mention the rockfall, even though it could not be safely sampled, to demonstrate you have completely considered all sources.

L106 : I'm not clear that you have answered the reviewer's question: did drying at 100 drive off any of the organics?

L107: both reviewers request information about the mass of cryoconite. I understand that you cannot include this information, but please ensure you include justification of this in the text (e.g. just as you have in your responses).

L212-230: ensure this detail is clear in your text

L293: would it be useful to include the correlation matrix in the main text? It sounds interesting, and might not get seen if relegated to the appendix!

L372: it might be worth mentioning this justification in the text in some way

Figure 1: can you ensure that there is some kind of overview of Sweden if appropriate? Not everyone knows where Tarfala is, and it's very useful for visualising proximity to sources

Fig. 1 and 6: perhaps a note in the caption about the braided stream?

*Reviewer 2*

Study site: both reviewers ask for detail about the snow. Your response is clear, but it would be useful to mention this in the text: if both reviewers ask, then your readers will too.

L108: see above. Some mention of relative mass of different cryoconite types would be worthwhile.

L269: impact of sunlight. This is indeed interesting, would you like to add a sentence highlighting it (and providing justification for some future work?)

 I look forward to reading the updated manuscript.

Best regards,

Dr Liz Bagshaw

TC Editor